# Early parafoveal semantic integration in natural reading

Yali Pan[1]*, Steven Frisson[1], Kara D Federmeier[2], Ole Jensen[1]

[1]Centre for Human Brain Health, School of Psychology, University of Birmingham, Birmingham, United Kingdom; [2]Department of Psychology, Program in Neuroscience, and the Beckman Institute for Advanced Science and Technology, University of Illinois, Champaign, United States

**Abstract** Humans can read and comprehend text rapidly, implying that readers might process multiple words per fixation. However, the extent to which parafoveal words are previewed and integrated into the evolving sentence context remains disputed. We investigated parafoveal processing during natural reading by recording brain activity and eye movements using MEG and an eye tracker while participants silently read one-line sentences. The sentences contained an unpredictable target word that was either congruent or incongruent with the sentence context. To measure parafoveal processing, we flickered the target words at 60 Hz and measured the resulting brain responses (i.e. *Rapid Invisible Frequency Tagging, RIFT*) during fixations on the pre-target words. Our results revealed a significantly weaker tagging response for target words that were incongruent with the previous context compared to congruent ones, even within 100ms of fixating the word immediately preceding the target. This reduction in the RIFT response was also found to be predictive of individual reading speed. We conclude that semantic information is not only extracted from the parafovea but can also be integrated with the previous context before the word is fixated. This early and extensive parafoveal processing supports the rapid word processing required for natural reading. Our study suggests that theoretical frameworks of natural reading should incorporate the concept of deep parafoveal processing.

**\*For correspondence:**
Y.Pan.1@bham.ac.uk

**Competing interest:** The authors declare that no competing interests exist.

## eLife assessment

This **important** study contributes to the understanding of how parafoveal words are neurally processed during naturalistic sentence reading. **Convincing** evidence is provided that the MEG response to a word can be modulated by the semantic congruency of a parafoveal target word. The study addresses a classic question in reading using a new Rapid Invisible Frequency Tagging (RIFT) technique, which can separately monitor the neural processing of multiple words during sentence reading.

## Introduction

Reading is a remarkable human skill that requires rapid processing of written words. We typically fixate each word for only 225–250ms, but nevertheless manage to encode its visual information, extract its meaning, and integrate it into the larger context, while also doing saccade planning (*Rayner, 2009*). To overcome the tight temporal constraints during reading, we preview the next word in the parafovea before moving our eyes to it (*Jensen et al., 2021*; *Reichle and Reingold, 2013*; *Schotter, 2018*). Substantial evidence suggests that parafoveal information can be extracted at various linguistic levels, including orthography (*Drieghe et al., 2005*; *Inhoff, 1989*; *Johnson et al., 2007*; *White, 2008*; *Williams et al., 2006*), phonology (*Ashby et al., 2006*; *Ashby and Rayner, 2004*; *Chace et al.,*

*2005*; *Miellet and Sparrow, 2004*; *Pollatsek et al., 1992*; *Rayner et al., 1995*), lexicality (*Kennedy and Pynte, 2005*; *Kliegl et al., 2006*), syntax (*Snell et al., 2017*; *Wen et al., 2019*), and semantics (*Rayner and Schotter, 2014*; *Schotter, 2013*; *Schotter et al., 2015*; *Schotter and Jia, 2016*); for a comprehensive review see *Schotter et al., 2012*. However, for semantics in particular, controversy remains about the extent and type of information extracted from parafoveal processing under various conditions. Moreover, it is unknown when and how the previewed semantic information can be used – i.e., integrated into the evolving sentence context – which is an integral component of the ongoing reading process.

For some time, it was claimed that parafoveal preview was limited to perceptual features of words and did not extend to semantics (*Inhoff, 1982*; *Inhoff and Rayner, 1980*; *Rayner et al., 2014*; *Rayner et al., 1986*). However, eye tracking-based evidence for the extraction of parafoveal semantic information began to emerge from studies that used languages other than English, including Chinese (*Tsai et al., 2012*; *Yan et al., 2012*; *Yan et al., 2009*; *Zhou et al., 2013*) and German (*Hohenstein et al., 2010*; *Hohenstein and Kliegl, 2014*), and was eventually extended into English (*Rayner and Schotter, 2014*; *Schotter et al., 2015*; *Schotter and Jia, 2016*; *Veldre and Andrews, 2018*; *Veldre and Andrews, 2017*; *Veldre and Andrews, 2016a*; *Veldre and Andrews, 2016b*). For example, (*Schotter and Jia, 2016*) showed preview benefits on early gaze measures for plausible compared to implausible words, even for plausible words that were unrelated to the target. These results demonstrate that semantic information can indeed be extracted from parafoveal words. However, due to the limitations of the boundary paradigm, which only assesses effects after target words have been fixated, it is challenging to precisely determine when and how parafoveal semantic processing takes place. Furthermore, it is generally hard to distinguish between the effects of cross-saccade integration (e.g. the mismatch between the preview and the word fixated) and the effects of how differing words fit into the context itself (*Veldre and Andrews, 2016a*; *Veldre and Andrews, 2016b*).

Complementary evidence showing that semantic information can be extracted parafoveally, even in English, comes from electrophysiological studies. Context-based facilitation of semantic processing can be observed as reductions in the amplitude of the N400 component (*Kutas and Hillyard, 1984*; *Kutas and Hillyard, 1980*), a negative-going event-related potential (ERP) response observed between about 300 and 500ms after stimulus onset, which has been linked to semantic access (*DeLong et al., 2014*; *Federmeier, 2022*; *Federmeier et al., 2007*; *Kutas and Federmeier, 2011*; *Lau et al., 2008*). Basic effects of contextual congruency on the N400 – smaller responses to words that do versus do not fit a sentence context (e.g. to 'butter' compared to 'socks' after 'He spread the warm bread with …') – are also observed for parafoveally-presented words (*Antúnez et al., 2022*; *Barber et al., 2013*; *Barber et al., 2010*; *López-Peréz et al., 2016*; *Meade et al., 2021*) and, even when all words are congruent, N400 responses to words in parafoveal preview, like those to foveated words, are graded by increasing context-based predictability (*Payne et al., 2019*; *Payne and Federmeier, 2017*; *Stites et al., 2017*). Although many of these effects have been measured in the context of unnatural reading paradigms (e.g. the 'RSVP flanker paradigm'), similar effects are obtain during natural reading. Using the stimuli and procedures from *Schotter and Jia, 2016*, *Antúnez et al., 2022* showed that N400 responses, measured relative to the fixation before the target words i.e., before the boundary change while the manipulated words were in parafoveal preview, were sensitive to the contextual plausibility of these previewed words. These studies suggest that semantic information is available from words before they are fixated, even if that information does not always have an impact on eye fixation patterns.

Thus, both eye tracking and electrophysiological studies have provided evidence suggesting that semantic information is extracted from words in parafoveal preview. However, most of these studies have been limited to measuring parafoveal preview from fixations to an immediately adjacent word, raising questions about exactly how far in advance semantic information might become available from parafoveal preview. Moreover, important questions remain about the extent to which parafoveally extracted semantic information can be functionally integrated into the building sentence-level representation. Although some ERP studies have found that the semantic information extracted from parafoveal preview is carried forward, affecting semantic processing when that same word is later fixated (*Barber et al., 2010*; *Payne et al., 2019*; *Stites et al., 2017*), other studies have not observed any downstream impact (*Barber et al., 2013*; *Li et al., 2015*). Furthermore, post-N400 ERP components, linked to more attentionally-demanding processes associated with message-building and revision, do

not seem to be elicited during parafoveal preview (*Li et al., 2023*; *Milligan et al., 2023*; *Payne et al., 2019*; *Schotter et al., 2023*). Therefore, critical questions remain about the time course and mechanisms by which semantic information is extracted and used during reading.

Answering those questions requires an approach that allows a more continuous and specific assessment of sensitivity to target word semantics during parafoveal processing across multiple fixations, and, in particular, that can speak to how attention is allocated across words during natural reading. We tackle these core issues using a new technique that combines the use of frequency tagging and the measurement of magnetoencephalography (MEG)-based signals.

Frequency tagging, also known as steady-state visually evoked potentials, involves flickering a visual stimulus at a specific frequency and then measuring the neuronal response associated with processing the stimulus (*Norcia et al., 2015*; *Vialatte et al., 2010*). It has been widely used to investigate visuospatial attention (*Gulbinaite et al., 2019*; *Kritzman et al., 2022*; *Müller et al., 2003*; *Müller et al., 1998*; *Norcia et al., 2015*; *Vialatte et al., 2010*) and has recently been applied to language processing (*Beyersmann et al., 2021*; *Montani et al., 2019*; *Wu et al., 2023*). However, the traditional frequency tagging technique flickers visual stimuli at a low-frequency band, usually below 30 Hz, such that the flickering can be visible and may interfere with the ongoing task. To address this limitation, we developed the rapid invisible frequency tagging (RIFT) technique, which involves flickering visual stimuli at a frequency above 60 Hz, making it invisible and non-disruptive to the ongoing task. Responses to RIFT have been shown to increase with the allocation of attention to the stimulus bearing the visual flicker (*Brickwedde et al., 2022*; *Drijvers et al., 2021*; *Duecker et al., 2021*; *Ferrante et al., 2023*; *Gutteling et al., 2022*; *Zhigalov et al., 2021*; *Zhigalov et al., 2019*; *Zhigalov and Jensen, 2022*; *Zhigalov and Jensen, 2020*). In our previous study, we adapted RIFT to a natural reading task and found temporally-precise evidence for parafoveal processing at the lexical level (*Pan et al., 2021*). The RIFT technique provides a notable advantage by generating a signal — the tagging response signal — specifically yoked to just the tagged word. This ensures a clear separation in processing the tagged word from the ongoing processing of other words, addressing a challenge faced by eye tracking and ERP/FRP approaches. Moreover, RIFT enables us to monitor the entire dynamics of attentional engagement with the tagged word, which may begin a few words before the tagged word is fixated.

In the current study, RIFT was utilised in a natural reading task to investigate parafoveal semantic integration. We recruited participants (n=34) to silently read one-line sentences while their eye movements and brain activity were recorded simultaneously by an eye-tracker and MEG. The target word in each sentence was always unpredictable (see Behavioural pre-tests in Methods) but was semantically congruent or incongruent with the preceding sentence context (for the characteristics of words, see *Table 1*). The target words were tagged by flickering an underlying patch, whose luminance kept changing in a 60 Hz sinusoid throughout the sentence presentation. The patch was perceived as grey, the same color as the background, making it invisible. To ensure that the flicker remained invisible across saccades, we applied a Gaussian transparent mask to smooth out sharp luminance changes around the edges (*Figure 1A*). Parafoveal processing of the target word was indexed by the RIFT responses recorded using MEG during fixations of pre-target words.

This paradigm allows us to address three questions. First, we aimed to measure when in the course of reading people begin to direct attention to parafoveal words. Second, we sought to ascertain when semantic information obtained through parafoveal preview is integrated into the sentence context in

**Table 1.** Characteristics of pre-target, target, and post-target words.

| | Pre-target | Target | Post-target |
|---|---|---|---|
| Word frequency | 124.5 (310.9) | 62.2 (77.0) | 3619.8 (6725.2) |
| Word Length | 5.5 (1.1) | 5.3 (1.1) | 5.4 (2.0) |
| Position in the sentence | 5.4 (3.0) | 6.4 (1.3) | 7.4 (1.3) |

Note. Word frequency is reported as the total CELEX frequency per million (*Davis, 2005*). Word length is the number of letters in a given word. Position in the sentence refers to the location in the sequence of words where a given word is presented. The number of words in each sentence is 11.6±1.7 (mean ± SD). All values shown here are mean with standard deviations in the parentheses. Note that the pre-target and post-target were identical for the congruent and incongruent conditions. The target word was counterbalanced over items such that it was congruent for one item and incongruent for another.

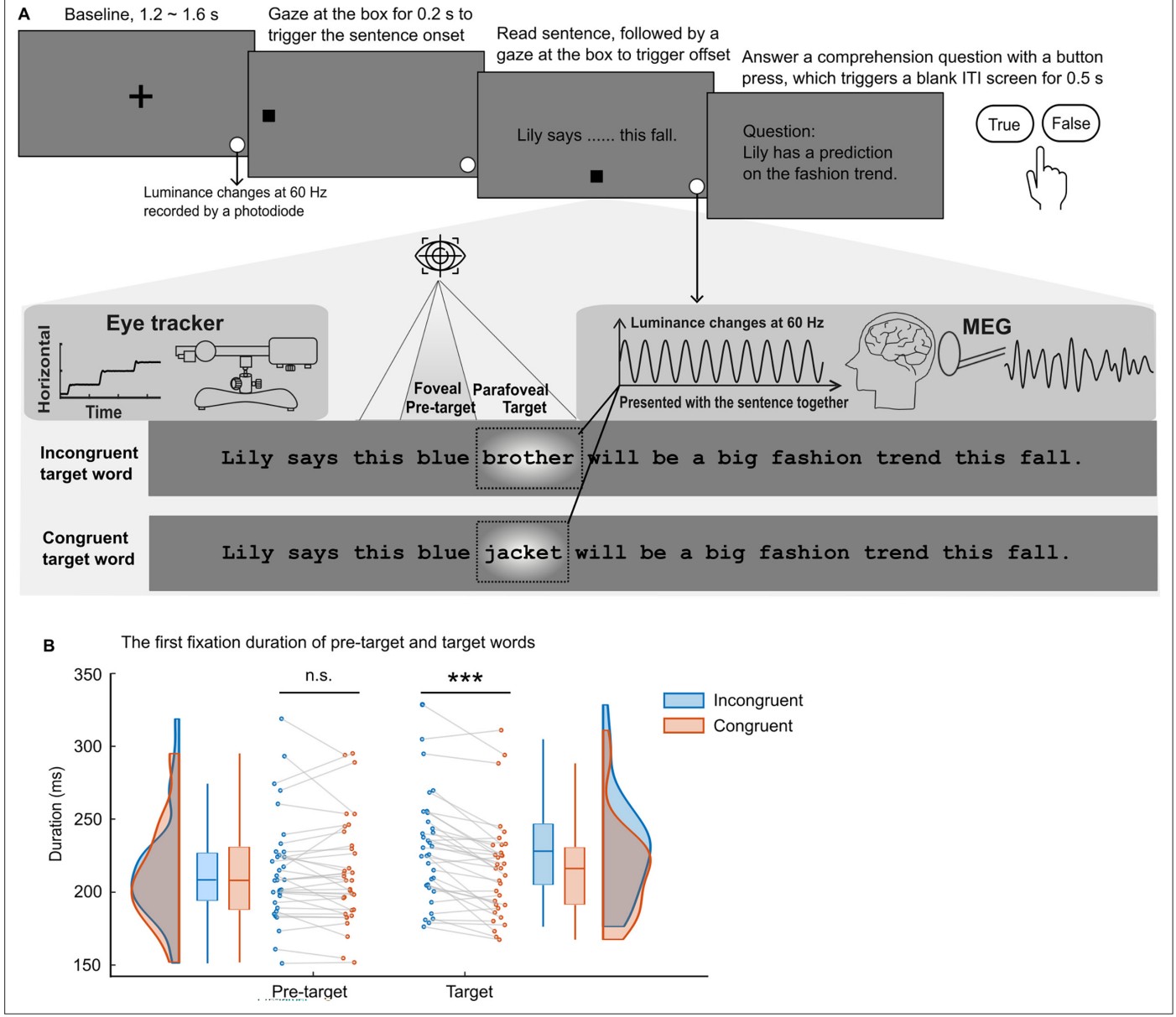

**Figure 1.** The paradigm and the eye movement metrics. (**A**) After the presentation of a cross-fixation at the screen centre for 1.2–1.6 s, a gaze-contingent box appeared near the left edge of the screen. Fixing the box for 0.2 s triggered the full sentence presentation. Participants (n=34) read 160 one-line sentences silently while brain activity and eye movements were recorded. Each sentence was embedded with one congruent or incongruent target word (see the dashed rectangle; not shown in the actual experiment). The target words could not be predicted based on the sentence context and word-level properties of congruent and incongruent targets were balanced by swapping them between two sentence frames. The target words were tagged by changing the luminance of the underlying patch (with a Gaussian mask) in a 60 Hz sinusoid throughout the sentence presentation (depicted as a bright blob, not shown in the actual experiment). Additionally, we included a small disk at the bottom right of the screen that displayed the tagging signal and was recorded by a photodiode throughout each trial. After reading, gazing at the bottom box for 0.2 s triggered the sentence offset. Twelve percent of the sentences were followed by a simple yes-or-no comprehension question. (**B**) The first fixation durations on the pre-target and target words when the target words were incongruent (in blue) or congruent (in orange) with the sentence context. Each dot indicates one participant (n=34). ***p<0.001; n.s., not statistically significant; ITI, inter-trial interval.

a manner that affects reading behaviours. Modulations of pre-target RIFT responses by the contextual congruity of target words would serve as evidence that parafoveal semantic information has not only been extracted and integrated into the sentence context but that it is affecting how readers allocate attention across the text. Third, we explored whether these parafoveal semantic attention effects have any relationship to reading speed.

# Results

## No evidence for semantic parafoveal processing in the eye movement data

Like prior work measuring eye fixations during English reading (*Inhoff, 1982*; *Inhoff and Rayner, 1980*; *Rayner et al., 2014*; *Rayner et al., 1986*), we found no evidence for parafoveal semantic processing in the eye movement data (*Figure 1B*, left). A paired t-test comparing first fixation durations on the pre-target word showed no effect of contextual (in)congruity ($t_{(33)}$ = 0.84, p=0.407, $d$=0.14, two-sided). However, first fixation durations on the target word were significantly longer when they were incongruent (versus congruent) with the context ($t_{(33)}$ = 5.99, p=9.83 × $10^{-7}$, $d$=1.03, two-sided pairwise t-test; *Figure 1B*, right). In addition, we found that the contextual congruity of target words affected later eye movement measures (i.e. total gaze duration and the likelihood of refixation after the first pass reading), with additional processing evident when the target words were incongruent with the context compared with when they were congruent (*Appendix 1—figure 1*).

## Parafoveal processing measured by RIFT

First, we performed a selection procedure to identify MEG sensors that responded to RIFT. We measured neural responses to the flickering target words by calculating the coherence between the MEG sensors and the tagging signal measured by a photodiode. A MEG sensor was considered a good tagging response sensor if it showed significantly stronger 60 Hz coherence during the pre-target intervals (with flicker) compared to the baseline intervals before the sentence presentation (without flicker). Both pre-target and baseline intervals were 1 s epochs. We then applied a cluster-based permutation test and identified sensor clusters that showed a robust tagging response ($p_{cluster}$ <.01; *Figure 2A*). Tagging response sensors were found in 29 out of 34 participants, and all subsequent analyses were based on these tagging response sensors (7.9±4.5 sensors per participant, M ± SD). The sources of these responses were localised to the left visual association cortex (Brodmann area 18; *Figure 2B*) using Dynamic Imaging Coherent Sources (DICS) (*Gross et al., 2001*).

Next, we characterised the temporal dynamics of attentional allocation to the flickering target word by calculating the 60 Hz coherence during fixations on several words surrounding the target word (*Figure 2C*). The resulting RIFT response curve revealed that significant attention was allocated to the target word as far as three words prior, spanning 15.3±2.7 letters (M ± SD), including the spaces between words. This range is consistent with previous estimations of the perceptual span of 12–15 letters during English reading (*McConkie and Rayner, 1975*; *Rayner, 2009*; *Rayner, 1975*; *Underwood and McConkie, 1985*), as reported in the eye movement literature. Moreover, as RIFT directly measures visual attention, the left-skewed RIFT response curve suggests that more visual attention is allocated towards the flickering target words before fixating on them, aligning with the left-to-right order of reading English. The normal size and left skewness of the perceptual span in our study suggest that RIFT did not influence attention distribution during natural reading. Notably, the strongest RIFT responses were observed during fixations on the pre-target word (i.e. word position N-1, *Figure 2C*), highlighting the suitability of RIFT for measuring neuronal activity associated with parafoveal processing during natural reading.

## Neural evidence for semantic parafoveal integration

Importantly, evidence for parafoveal semantic integration was found using RIFT (*Figure 3*). The pre-target coherence was weaker when the sentence contained a contextually incongruent word, compared to when it was congruent (*Figure 3A*). We conducted a pairwise t-test and found a significant effect on the averaged pre-target coherence at 60 Hz ($t_{(28)}$ = –2.561, p=0.016, $d$=0.476, two-sided pairwise; *Figure 3B*). To avoid any contamination of the parafoveal measure with activity from target fixation, pre-target coherence was averaged over the minimum pre-target fixation duration across both conditions for each participant (97.4±14.1 ms, M ± SD, denoted as a dashed rectangle). Next, we conducted a jackknife-based latency estimation and found that the congruency effect on the 60 Hz pre-target coherence had a significantly later onset when previewing an incongruent (116.0±1.9 ms, M ± SD) compared to a congruent target word (91.4±2.1 ms, M ± SD, denoted as a dashed rectangle; $t_{(28)}$ = –2.172, p=0.039, two-sided; *Figure 3C*). Therefore, both the magnitude and onset latency of the pre-target coherence were modulated by the contextual congruency of the target word, providing

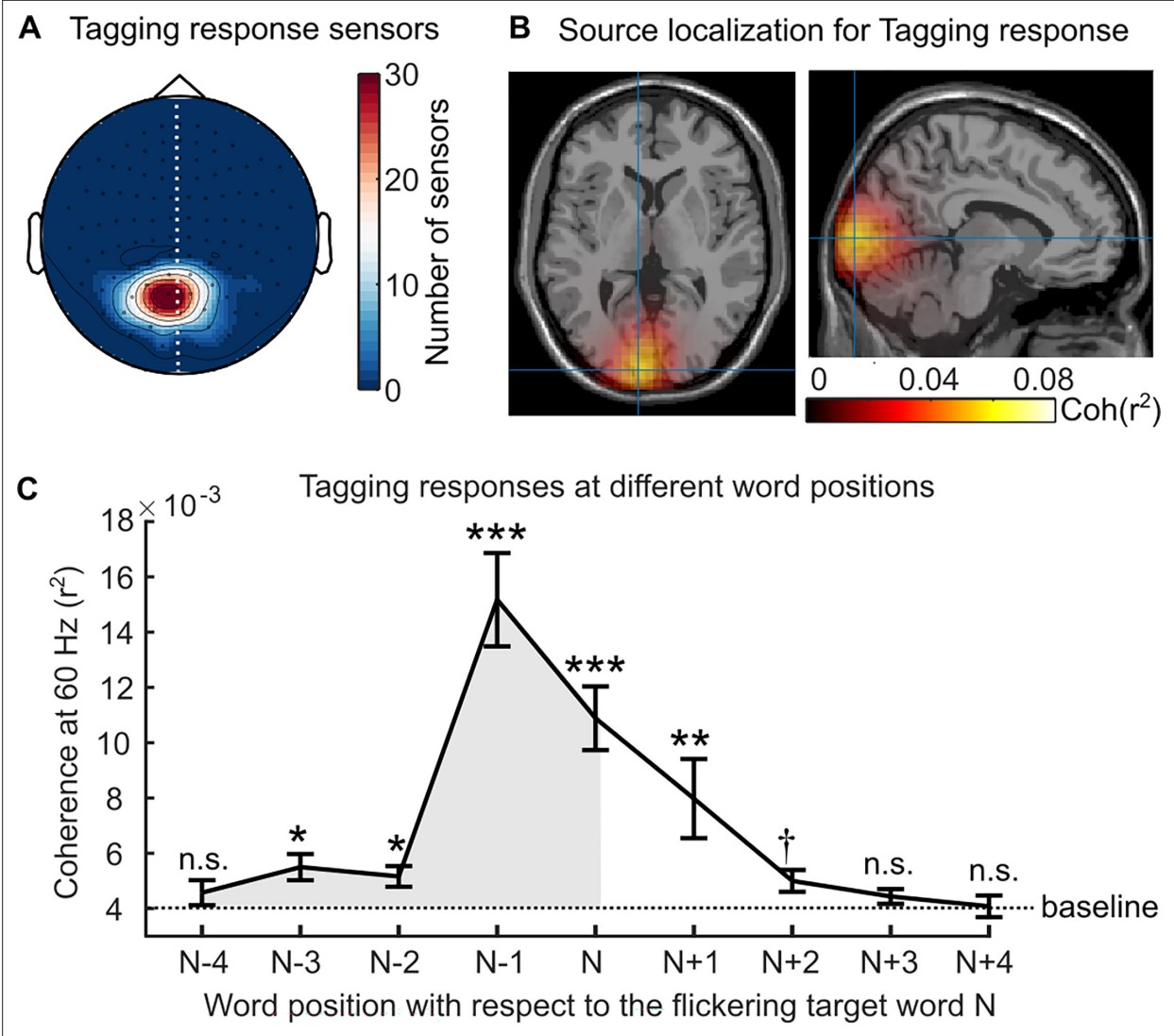

**Figure 2.** Neural responses to the rapid invisible frequency tagging (RIFT). (**A**) Topography of the RIFT response sensors over all participants (7.9±4.5 sensors per participant, M ± SD). These sensors showed significantly stronger coherence to the tagging signal during the pre-target interval (with target words flickering in the parafovea) compared with the baseline interval (no flicker). Further analyses only included participants who had a RIFT response (n=29). (**B**) The source of the RIFT response sensors was localised to the left visual association cortex (MNI coordinates [-9–97 3] mm, Brodmann area 18). (**C**) The averaged 60 Hz coherence over the RIFT response sensors when participants fixated on words at different positions, where n indicates the target word and n-1 indicates the pre-target word. Error bars indicate the SE over participants (n=29). The shaded area indicates the RIFT responses when previewing the flickering target words. We compared the RIFT response at each word position with the baseline (the dashed line). ***p<0.001, **p<0.01, *p<0.05, †p=0.051; n.s., not statistically significant.

neural evidence that semantic information is integrated into the context during parafoveal processing, detectable within 100 ms after readers fixate on the pre-target word.

We conducted a similar analysis of the coherence measured when participants fixated on the target word and found no significant modulations related to the contextual congruity of that target word, in either the magnitude ($t_{(28)}$ = 0.499, p=0.622, $d$=0.093, two-sided pairwise) or onset latency ($t_{(28)}$ = −0.280, p=0.782); (**Figure 4**) of the RIFT response. Thus, the parafoveal semantic integration effect identified during the pre-target intervals cannot be attributed to signal contamination from fixations on the target word induced by the temporal smoothing of filters.

## Parafoveal semantic integration is related to individual reading speed

The RIFT effects of congruency observed during the parafoveal preview of the targets showed that readers tend to allocate less attention to upcoming text when an upcoming word is semantically

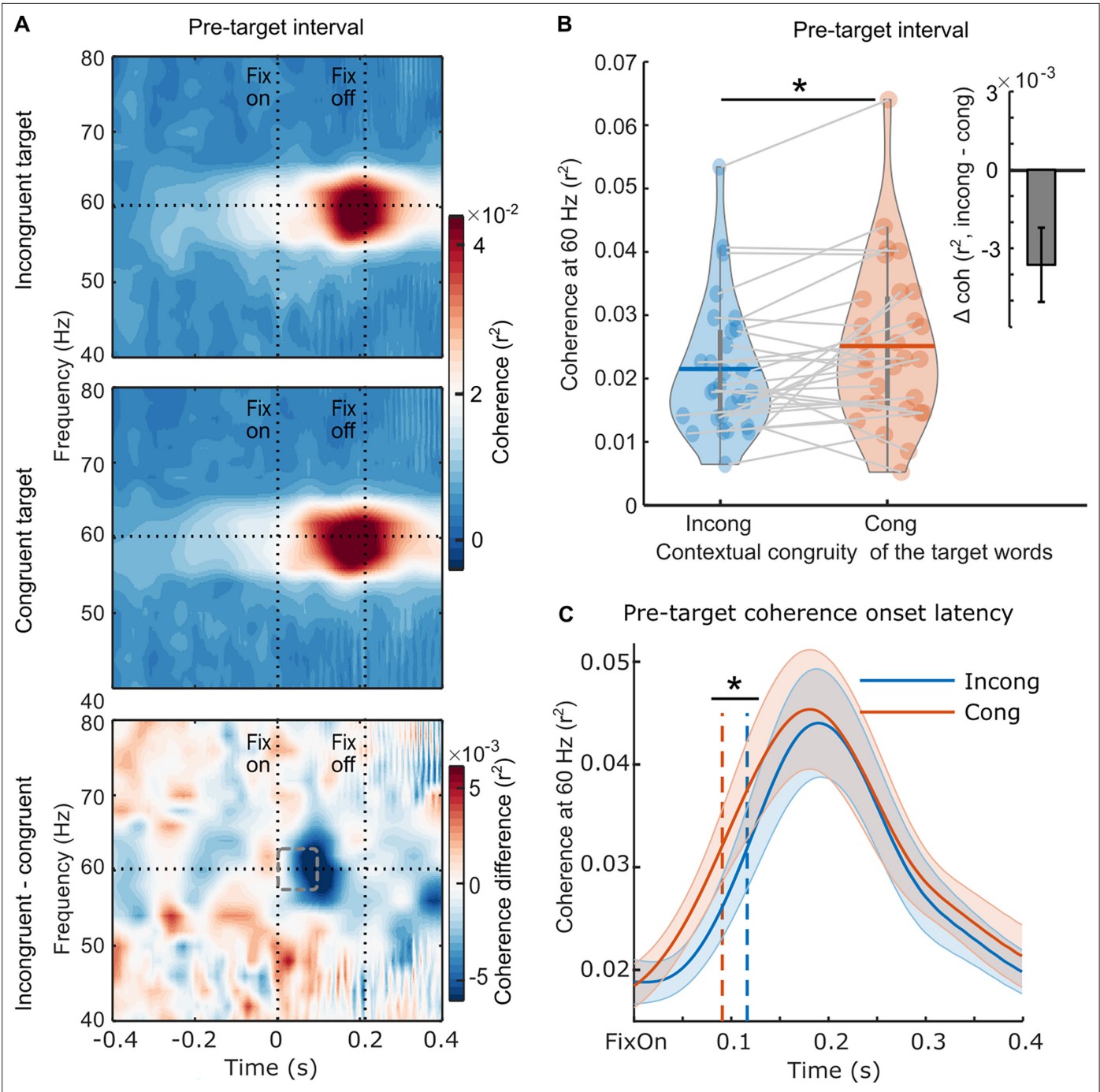

**Figure 3.** Neural evidence for parafoveal semantic integration. (**A**) The pre-target coherence spectrum averaged over the rapid invisible frequency tagging (RIFT) response sensors at the group level (n=29) when the subsequent target words were incongruent with the sentence context (top panel), congruent with the sentence context (middle panel), and the difference between the two conditions (bottom panel). The horizontal line indicates the tagging frequency at 60 Hz. The two vertical lines indicate the first fixation onset of the pre-target words and the average fixation offset. (**B**) The averaged 60 Hz coherence during the minimum pre-target intervals for each participant (97.4±14.1ms, M ± SD; denoted as a dashed rectangle) with respect to the incongruent and congruent target words. Each dot indicates one participant, the horizontal lines inside of the violins indicate the mean values. The upright inserted figure shows the pre-target coherence difference over participants with the error bar as SE. (**C**) The onset latency of the pre-target coherence at the group level (n=29). The onset latency refers to the time when the coherence curve reaches its half maximum, denoted by the dotted lines. Zero time-point indicates the first fixation onset of the pre-target words. The shaded area shows SE around the mean value. *p<.05.

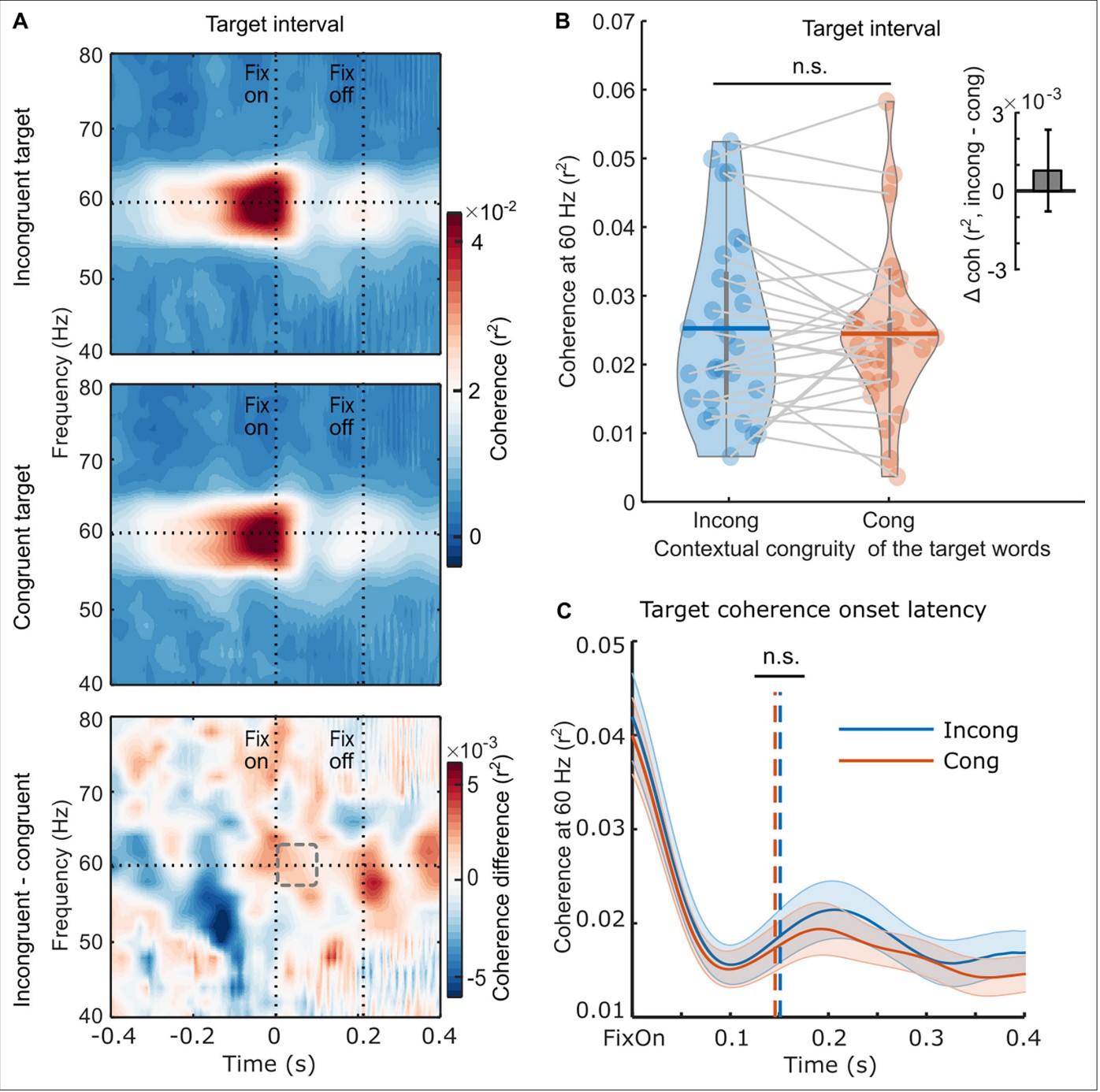

**Figure 4.** Neural responses to the rapid invisible frequency tagging (RIFT) during the target interval. (**A**) The target coherence spectrum averaged over the RIFT response sensors at the group level (n=29) when the target words were incongruent with the sentence context (top panel), or congruent with the sentence context (middle panel); the bottom panel shows the difference between the two conditions. The horizontal line indicates the tagging frequency at 60 Hz. The two vertical lines indicate the first fixation onset of the target words and the averaged fixation offset. (**B**) We averaged the 60 Hz coherence within the minimum target fixation duration over participants (97.6±15.7 ms, M ± SD; denoted as a dashed rectangle). Each dot indicates one participant, and the horizontal lines inside of the violins indicate the mean values. The upright inserted figure shows the target coherence difference over participants with the error bar as SE. (**C**) A jackknife-based method was used to calculate the onset latency of the average coherence at the group level. The onset latency refers to the time when the coherence curve reaches its half maximum, denoted by the dotted lines. n.s., not statistically significant.

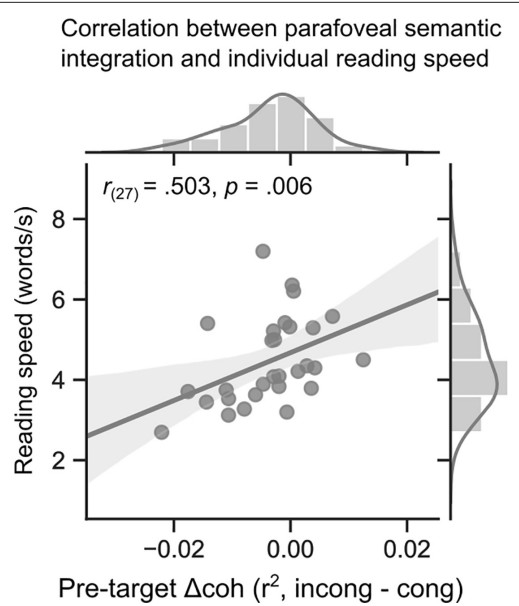

**Figure 5.** Individual reading speed is correlated with the magnitude of the rapid invisible frequency tagging (RIFT) congruency effect. Reading speed was measured as the number of words read per second in the congruent sentences. The RIFT effect was measured as the coherence difference during the pre-target fixations for sentences containing incongruent and congruent target words. Each dot indicates one participant (n=29, Spearman correlation). The shaded area represents the 95% CI.

incongruent compared to when all words are congruent. If readers differ in the extent to which their attention is 'repelled' by incongruent words, then we might expect that the magnitude of the RIFT effect would be related to reading speed. Therefore, we conducted a correlation analysis to investigate this relationship (**Figure 5**). Individual reading speed was quantified as the number of words read per second from the congruent sentences in the study. We found a positive correlation between the pre-target coherence difference (incongruent - congruent) and individual reading speed ($r_{(27)}$=0.503, p=0.006; Spearman's correlation). This suggests that readers who show greater shifts in attentional allocation in response to semantic incongruity read more slowly on average.

## Discussion

In the current natural reading study, we utilised RIFT to probe for evidence that readers are sensitive to the effect of contextual congruity of an upcoming target word during parafoveal processing. We found no significant modulation of fixation durations of pre-target words based on the contextual congruity of the target word (**Figure 1B**). However, we observed a significant difference in the amount of covert attention allocated to the target when previewing congruent and incongruent target words (**Figure 3**). Specifically, we found lower RIFT responses for parafoveal words that were incongruent compared to congruent with the previous context. Because the target words were always of low predictability, their semantic congruence could only be appreciated if they had been integrated (to some extent) with the unfolding context. Thus, the RIFT patterns provide compelling neural evidence that semantic information can not only be extracted but also integrated during parafoveal processing.

More specifically, we observed that pre-target coherence was weaker in magnitude (**Figure 3B**) and had a later onset latency (**Figure 3C**) in response to a contextually incongruent target word compared to a congruent one. Two possible explanations for these findings can be considered. First, the decreased RIFT responses may be due to changes in the pattern of allocation of attention across the text during reading. When reading in English, attention continuously shifts from left to right. If the semantic information previewed in the parafovea cannot be easily integrated into the context, this pattern may be interrupted, leading to delayed and/or reduced allocation of attention to the parafoveal word, possibly because readers shift more attention to the currently fixated word or to previous words to ensure that they have decoded and understood what they have read thus far. On this view, the RIFT finding may reflect a covert 'regression' of attention, similar to overt eye-movement regressions that sometimes occur when readers encounter semantically incongruous words (**Antúnez et al., 2022**; **Braze et al., 2002**; **Ni et al., 1998**; **Rayner et al., 2004**) (also see **Appendix 1—figure 1A**). Alternatively, the reduction in RIFT responses could arise if readers shift attentional resources away from the text altogether. Previous work has demonstrated that tagging responses decrease as attention shifts from an external task (e.g. counting visual targets) to an internal task (e.g. counting heartbeats) (**Kritzman et al., 2022**). Similarly, in a reading scenario, visually perceiving the flickering word constitutes an external task, while the internal task involves the semantic integration of previewed information into the context. If more attentional resources are internally directed when faced with the challenge of integrating a contextually incongruent word, fewer attentional resources would

remain for processing the flickering word. This may be the kind of shift reflected in the reduction in RIFT responses. On either account, the reduced forward allocation of attention diminishes parafoveal processing, and, in turn, may tend to slow reading speed, as supported by our correlation results (*Figure 5*).

Our results also provide information about the time course of semantic integration, as we found evidence that readers appreciated the incongruity – and thus must have begun to integrate the semantics of the parafoveal words with their ongoing message-level representation – by as early as within 100ms after fixating on the pre-target word. The timing of this parafoveal semantic effect appears remarkably early, considering that typical semantic access for a single word occurs no earlier than around 200ms, as demonstrated in the visual word recognition literature (*Carreiras et al., 2014*). For instance, in a Go/NoGo paradigm, the earliest distinguishable brain activity related to category-related semantic information of a word occurs at 160ms (*Amsel et al., 2013*; *Hauk et al., 2012*). Therefore, the RIFT results presented here suggest that natural reading involves parallel processing that spans multiple words. The level of (covert) attention allocated to the target word, as indexed by the significant difference in RIFT responses compared to the baseline interval, was observed even three words in advance (see *Figure 2C*). This initial increase in RIFT coincided with the target entering the perceptual span (*McConkie and Rayner, 1975*; *Rayner, 1975*; *Underwood and McConkie, 1985*), likely aligning with the initial extraction of lower-level perceptual information about the target. The emerging sensitivity of the RIFT signal to target plausibility, detected around 100ms after the fixation on the pre-target word, suggests that readers at that time had accumulated sufficient semantic information about the target words and integrated that information with the evolving context. Therefore, it is plausible that the initial semantic processing of the target word commenced even before the pre-target fixation and was distributed across multiple words. This parallel processing of multiple words facilitates rapid and fluent reading.

Our findings have significant implications for theories of reading. The occurrence and early onset of semantic integration in parafoveal vision suggest that words are processed in an exceptionally parallel manner, posing a challenge for existing serial processing models (*Reichle et al., 2009*; *Reichle et al., 2006*; *Reichle et al., 2003*; *Reichle et al., 1998*). At the same time, it is important to note that the fact that semantic integration begins in parafoveal vision does not mean that it is necessarily completed before a word is fixated. The fact that we observed semantic congruency effects on the fixation durations of the target words (*Figure 1B*) suggests that additional processing is required to fully integrate the semantics with overt attention in foveal vision. This also aligns with previous studies that found some ERP responses to semantic violations, including the LPC (Late Positive Component), are elicited only during foveal processing, but not during parafoveal processing (*Li et al., 2023*; *Milligan et al., 2023*; *Payne et al., 2019*; *Schotter et al., 2023*).

Thus, RIFT measures complement eye tracking (and other) measures, providing unique information revealing multiple mechanisms at work during natural reading. The results of the present study are aligned with the SWIFT model of eye movement control in natural reading (*Engbert et al., 2005*), wherein the activation field linked to a given word is hypothesised to be both temporally and spatially distributed. Indeed, we found that the initial increase in covert attention to the target word occurred as early as three words before, as measured by RIFT responses (*Figure 2C*). These covert processes enable the detection of semantic incongruity (*Figure 3B* and *Figure 3C*). However, it may occur at the non-labile stage of saccade programming, preventing its manifestation in fixation measures of the currently fixated pre-target word (*Figure 1B*). Therefore, the RIFT technique's capacity to yoke patterns to a specific word offers a unique opportunity to track the activation field of word processing during natural reading. Additional processes, which do impact overt eye movement patterns, are then brought to bear when the target words are fixated, resulting in increased fixation durations for incongruous words. At that same point (i.e. the target word), however, the RIFT responses showed a null effect of congruency (*Figure 4*); it may be that the RIFT technique is better suited to capturing parafoveal compared to foveal attentional processes, in part because there are more motion-sensitive rod cells in the parafoveal than foveal area. Finally, even after readers move away from fixating the word, attention to the target can persist or be reinstated, as evidenced by patterns of regressions (*Appendix 1—figure 1A*). Therefore, during natural reading, attention is distributed across multiple words. The highly flexible and distributed allocation of attention allows readers to be parallel processors and thereby read fluently and effectively (*Engbert et al., 2005*; *Engbert et al., 2002*; *Snell et al.,*

2018; *Snell and Grainger, 2019*). Our natural reading paradigm, where all words are available on the screen and saccadic eye movements are allowed, makes it possible to capture the extensive parallel processing. Moreover, saccades have been found to coordinate our visual and oculomotor systems, further supporting the parallel processing of multiple words during natural reading (*Pan et al., 2023*).

Two noteworthy limitations exist in the current study. Firstly, the construction of pretarget–target word pairs consistently follows an adjective-noun phrase structure, potentially leading to semantic violations arising from immediate local incongruence rather than a broader incongruence derived from the entire sentential context. While the context preceding target words was deliberately minimised to ensure a pure effect of bottom-up parafoveal processing rather than the confounding impact of top-down prediction, it is essential to recognize that information from both local and global contexts can exert distinct effects on word processing during natural reading (*Wong et al., 2024*). Future investigations should incorporate more information-rich contexts to explore the extent to which the parafoveal semantic integration effect observed in this study can be generalised. Second, the correlation analysis between the pre-target RIFT effect and individual reading speed (*Figure 5*) does not establish a causal relationship between parafoveal semantic integration and reading performance. Given that the comprehension questions in the current study were designed primarily to maintain readers' attention and the behavioural performance reached a ceiling level, employing more intricate comprehension questions in future studies would be ideal to accurately measure reading comprehension and reveal the impact of semantic parafoveal processing on it.

In summary, our findings show that parafoveal processing is not limited to simply extracting word information, such as lexical features, as demonstrated in our previous study (*Pan et al., 2021*). Instead, the previewed parafoveal information from a given word can begin to be integrated into the unfolding sentence representation well before that word is fixated. Moreover, the impact of that parafoveal integration further interacts with reading comprehension by shaping the time course and distribution of attentional allocation – i.e., by causing readers to move attention away from upcoming words that are semantically incongruous. These results support the idea that words are processed in parallel and suggest that early and deep parafoveal processing may be important for fluent reading.

## Materials and methods
### Participants
We recruited 36 native English speakers (24 females, 22.5±2.8 years old, mean ± SD) with normal or corrected-to-normal vision. All participants are right-handed and without any history of neurological problems or a language disorder diagnosis. Two participants were excluded from the analysis due to poor eye tracking or falling asleep during the recordings, which left 34 participants (23 females). This sample size was determined based on our previous study with a similar experimental design (*Pan et al., 2021*). The study was approved by the University of Birmingham Ethics Committee (under the approved Programme ERN_18-0226AP27). The informed consent form was signed by all participants after the nature and possible consequences of the studies were explained. Participants received £15 per hour or course credits as compensation for their participation.

### Stimuli
In total participants read 277 sentences, of which 117 sentences were fillers from a published paper (*White, 2008*). The filler sentences were all plausible and were included to make sure the incongruent sentences were less than one-third of the sentence set. We constructed the remaining 160 sentences with 80 pairs of target words. In all sentences the context was low constraint; i.e., none of the target words could be predicted by the prior context (see Behavioural pre-tests below for details). The target word in each sentence was either incongruent or congruent with the sentence. To focus on semantic integration and avoid any confounds of word-level properties, we embedded each pair of target words in two different sentence frames. By swapping the target words within a pair of sentences, we created four sentences: two congruent ones and two incongruent ones. These were then counterbalanced over participants. In this way, we counterbalanced across lexical characteristics of the target words and characteristics of the sentence frames within each pair. Each participants read one version of the sentence set (A or B). For example, for the target pair *brother/jacket*, one participant read them in the congruent condition in the sentence set version A; while another participant read them in the

incongruent condition in version B (see below, targets are in italic type for illustration, but in normal type in the real experiment).

   A. Last night, my lazy *brother* came to the party 1 min before it was over.
   Lily says this blue *jacket* will be a big fashion trend this fall.
   B. Last night, my lazy *jacket* came to the party 1 min before it was over.
   Lily says this blue *brother* will be a big fashion trend this fall.

   For all sentences, the pre-target words were adjectives, and the target words were nouns (for detailed characteristics of the words please see *Table 1*). The word length of pre-target words was from 4 to 8 letters, and for target words was from 4 to 7 letters. The sentences were no longer than 15 words or 85 letters. The target words were embedded somewhere in the middle of each sentence and were never the first three or the last three words in a sentence. Please see Appendix for the full list of the sentence sets that were used in the current study.

## Behavioural pre-tests

We recruited native English speakers for two behavioural pre-tests of the sentence sets. These participants did not participate in the MEG session.

### Predictability of target words

We carried out a cloze test to estimate the predictability of the target words and the contextual constraint of the sentences. Participants read sentence fragments consisting of the experimental materials up to but not including the target words. Then participants were asked to write down the first word that came to mind that could continue the sentence (no need to complete the whole sentence). Example:

   Last night, my lazy _______________
   Lily says this blue _______________

   The predictability of a word was estimated as the percentage of participants who wrote down exactly this word in the cloze test. A target word with less than 10% predictability was deemed to be not predicted by the sentence context. In addition, sentences for which no word was predicted with 50% or greater probability were a low constraint. Twenty participants (aix males, 24.2±2.0 years old, mean ± SD) took part in the first round of the pre-test. Eight sentences were replaced with new sentences because the target words were too predictable and/or the sentence was too constraining. We then conducted a second round of the predictability test with 21 new participants (seven males, 25.0±6.0 years old). None of the target words in this final set were predictable (2.3%±4.8%, mean ± SD), and all the sentence contexts were low constraint (25.2%±11.8%).

### Plausibility of sentences

Two groups of participants were instructed to rate how plausible (or acceptable) each sentence was in the sentence set version A or B separately. Plausibility was rated on a seven-point scale with plausibility increasing from point 1–7. Sentences in the experiment were designed to be either highly implausible (the incongruent condition) or highly plausible (the congruent condition). To occupy the full range of the scale, we constructed 70 filler sentences with middle plausibility (e.g. sentence 1 below). In this example, sentences 2 and 3 were the incongruent and congruent sentences from the experiment.

| | Implausible | | | | | | Plausible |
|---|---|---|---|---|---|---|---|
| 1. Kate said that she saw lots of stars twinkling in the sky at noon. | 1 | 2 | 3 | 4 | 5 | 6 | 7 |
| 2. Lily says this blue brother will be a big fashion trend this fall. | 1 | 2 | 3 | 4 | 5 | 6 | 7 |
| 3. Little Jimmy picked up a box and put some coins inside of it. | 1 | 2 | 3 | 4 | 5 | 6 | 7 |
| ...... | 1 | 2 | 3 | 4 | 5 | 6 | 7 |

   For version A we recruited 27 participants (four males, 22.8±6.1 years old, mean ± SD): The plausibility rating for the incongruent sentences was 2.08±0.79 (mean ± SD); while for the congruent sentences was 6.18±0.56. For sentence set version B we recruited 22 participants (four males, 21.1±2.3 years old, one invalid dataset due to incomplete responses): The plausibility rating was 1.81±0.41 (mean ± SD) for the sentences in the incongruent condition and 6.15±0.47 for the sentences

in the congruent condition. These results showed that in both versions of the sentences set, incongruent sentences were viewed as highly implausible and congruent sentences as highly plausible.

## Experimental procedure

Participants were seated 145 cm away from the projection screen in a dimly lit magnetically shielded room. The MEG gantry was set at 60 degrees upright and covered the participant's whole head. We programmed in Psychophysics Toolbox –3 (*Kleiner et al., 2007*) to present the one-line sentences on a middle-grey screen (RGB [128 128 128]). All words were displayed in black (RGB [0 0 0]) with an equal-spaced Courier New font. The font size was 20 and the font type was bold so that each letter and space occupied 0.316 visual degrees. The visual angle of the whole sentence was no longer than 27 visual degrees in the horizontal direction. The sentence set was divided into five blocks, each of which took about 7 min. There was a break of at least 1 min between blocks and participants pressed a button to continue the experiment at any time afterwards. Participants were instructed to read each sentence silently at their own pace and to keep their heads and body as stable as possible during the MEG session. Eye movements were acquired during the whole session. In total, the experiment took no longer than 55 min. While the current study was conducted using MEG, these procedures might also work with EEG. If so, this would make our approach accessible to more laboratories as EEG is less expensive. However, there are currently no studies directly comparing the RIFT response in EEG versus MEG. Therefore, it would be of great interest to investigate if the current findings can be replicated using EEG.

Within a trial, there was first a fixation cross presented at the centre of a middle-grey screen for 1.2–1.6 s. This was followed by a black square with a radius of 1 degree of visual angle. This square was placed at the vertical center, 2 degrees of visual angle away from the left edge of the screen. Participants had to gaze at this black 'starting square' for at least 0.2 s to trigger the onset of the sentence presentation. Afterward, the sentence would start from the location of the square (*Figure 1A*). The sentence was presented with an 'ending square' 5 degrees of visual angle below the screen centre. The 'ending square' was the same size as the 'starting square' but in grey colour (RGB [64 64 64]). A gaze at this 'ending square' for at least 0.1 s would end the presentation of the sentence. Then the trial ended with a blank middle-grey screen that lasted for 0.5 s. Randomly, 12% of the trials were followed by a statement about the content of the sentence that was just presented, and participants needed to answer 'True or False' by pressing a button. For example, the statement for sentence 2 was 'Lily has a prediction about the fashion trend in this fall,' and the correct answer was 'True.' The statement for sentence 3 was 'Little Jimmy didn't have a box,' and the correct answer was 'False.' All participants read the sentences carefully as shown by the high accuracy of answering (96.3%±4.7%, mean ± SD).

## RIFT

### Projection of the sentence stimuli

We projected the sentences from the stimulus computer screen in the experimenter room to the projection screen inside of the MEG room using a PROPixx DLP LED projector (VPixx Technologies Inc, Canada). The refresh rate of the PROPixx projector was up to 1440 Hz, while the refresh rate of the stimulus screen was only 120 Hz (1920×1200 pixels resolution). We displayed the sentence repeatedly in four quadrants of the stimulus computer screen. In each quadrant, the words were coded in three colour channels as RGB. The projector then interpreted these 12 colour channels (three channels×four quadrants) as 12 individual greyscale frames, which were projected onto the projection screen in rapid succession. Therefore, the projection screen refreshed at 12 times the rate of the stimulus computer screen.

### Flickering of the target word

We added a square patch underneath the target word to frequency tag the target word. The side length of the square patch was the width of the target word plus the spaces on both sides (2–3° visual angle). We flickered the patch by changing its luminance from black to white at a 60 Hz sinusoid (*Figure 1A*). To reduce the visibility of the patch edges across saccades, we applied a Gaussian smoothed transparent mask on top of the square patch. The mask was created by a two-dimensional Gaussian function (*Equation 1*):

$$mask = exp\left(-\frac{x^2}{2\sigma^2} - \frac{y^2}{2\sigma^2}\right)$$

where, $x$ and $y$ are the mesh grid coordinates for the flickering patch, and $\sigma$ is the $x$ and $y$ spread of the mask with $\sigma$=0.02 degrees.

On average, the patch was perceived as middle-grey, the same colour as the background screen, which made it invisible to participants. The target word was still black, the same colour as the other words on the screen. To record the tagging signal, we attached a custom-made photodiode (Aalto NeuroImaging Centre, Finland) to the disk at the bottom right corner of the screen. The luminance of the disk varied the same as that of the flickering patch underneath the target word. The photodiode was plugged into the MEG system as an external channel.

## Data acquisition

### MEG

Brain data were acquired with a 306-sensor TRIUX Elekta Neuromag system, which consisted of 204 orthogonal planar gradiometers and 102 magnetometers (Elekta, Finland). After participants signed the consent form, we attached four head-position indicator coils (HPI coils) to their heads: two on the left and right mastoid bone, and two on the forehead with at least 3 cm distance in between. After-ward, we used a Polhemus Fastrack electromagnetic digitizer system (Polhemus Inc, USA) to digitize the locations for three bony fiducial points: the nasion, left, and right preauricular points. Then we digitised the four HPI coils. Furthermore, at least 200 extra points were acquired, which were distrib-uted evenly and covered the whole scalp. These points were used later in the source analysis when spatially co-registering the MEG head model with individual structural MRI images. The sampling rate of the MEG system was 1000 Hz. Data were band-pass filtered prior to sampling from 0.1 to 330 Hz to reduce aliasing effects.

### Eye movements

We used an EyeLink 1000 Plus eye-tracker (long-range mount, SR Research Ltd, Canada) to track eye movements throughout the whole MEG session. The eye tracker was placed on a wooden table in front of the projection screen. The centre of the eye tracker was at the middle line of the projection screen, and the top of the eye tracker reached the bottom edge of the screen. The distance between the eye-tracker camera and the centre of the participant's eyes was 90 cm. We recorded the hori-zontal and vertical positions as well as the pupil size from the left eye, at a sampling rate of 1000 Hz. Each session began with a nine-point calibration and validation test. The test was accepted if the eye-tracking error was below 1 visual degree both horizontally and vertically. During the session, we performed a one-point drift-checking test every three trials and after the break between blocks. If the drift checking failed or the sentence presentation was unable to be triggered through gazing, a nine-point calibration and validation test was conducted again.

### MRI

After MEG data acquisition, participants were asked to come to the laboratory another day to have an MRI image acquired. We acquired the T1-weighted structural MRI image using a 3-Tesla Siemens PRISMA scanner (TR = 2000 ms, TE = 2.01 ms, TI = 880 ms, flip angle = 8 degrees, FOV = 256×256×208 mm, 1 mm isotropic voxel). For 11 participants who dropped out of the MRI acquisition, the MNI template brain (Montreal, Quebec, Canada) was used instead in the source analysis later.

## Eye movement data analysis

We extracted the fixation onset events from the EyeLink output file. The EyeLink parsed fixation events based on the online detection of saccade onset using the following parameters: the motion threshold as 0.1 degrees, the velocity threshold as 30 degrees/s, and the acceleration threshold as 8000 degrees/sec2. These conservative settings were suggested by the EyeLink user manual for reading studies, as they can prevent false saccade reports and reduce the number of micro-saccades, and lengthen fixation durations.

Only the fixation that first landed on a given word was selected. The first fixation durations were averaged within the incongruent and congruent conditions for pre-target and target words. Pairwise,

two-sided *t*-test were conducted on the first fixation durations of pre-target and target words separately (conducted in R *R Development Core Team, 2013*). In addition to the early eye movement measure of the first fixation duration, we also conducted *t*-tests for two later eye movement measures. The likelihood of refixation was measured as the proportion of trials on which there was at least one saccade that regressed back to that word. The total gaze duration was the sum of all fixations on a given word, including those fixations during regression or re-reading.

## MEG data analyses

The data analyses were performed in MATLAB R2020a (Mathworks Inc, USA) by using the FieldTrip (*Oostenveld et al., 2011*) toolbox (version 20200220), following the FLUX MEG analysis pipeline (*Ferrante et al., 2022*), and custom-made scripts (see Code availability for the shared link).

### Pre-processing

We first band-pass filtered the MEG data from 0.5 to 100 Hz using phase-preserving two-pass Butterworth filters. Subsequently, detrending was applied individually to each channel of the continuous raw data to factor out the linear trend. Malfunctioning sensors were removed based on inspecting the data quality during online recording (0–2 sensors per participant). Afterward, the data were decomposed into independent components using an independent component analysis (ICA) (*Ikeda and Toyama, 2000*). The number of components was the same as the number of good MEG sensors in the dataset (306 or less). We only removed bad components that related to eye blinks, eye movements, and heartbeat by visually inspecting the components (3.4±0.7 components per participant, M ± SD, range from 2 to 5 components).

MEG segments were extracted from –0.5–0.5 s intervals aligned with the first fixation onset of the pre-target and target words, respectively (see Eye movement data analysis, above, for information on how fixation onsets were defined). Segments with fixation durations shorter than 0.08 s or longer than 1 s were discarded. We also extracted 1 s long baseline segments, which were aligned with the cross-fixation onset before the sentence presentation. We manually inspected all segments to further identify and remove segments that were contaminated by muscle or movement artefacts.

### Coherence calculation

We calculated the coherence between the MEG sensors and the photodiode (i.e. the tagging signal) to quantify the tagging responses. The amplitude of the photodiode channel was normalised across each segment. To estimate the coherence spectrum in the frequency domain over time, we filtered the segments using hamming tapered Butterworth bandpass filters (fourth order, phase preserving, two-pass). The frequency of interest was from 40 to 80 Hz in a step of 2 Hz. For each center frequency point, the spectral smoothing was ±5 Hz. For example, the filter frequency range for 60 Hz was from 55 to 65 Hz. We performed a Hilbert transform to obtain the analytic signals for each centre frequency point, which then were used to estimate the coherence (*Equation 2*):

$$coh\left(t\right) = \frac{\left| \frac{1}{n} \sum_{j=1}^{n} m_{x_j}\left(t\right) m_{y_j}\left(t\right) e^{i\phi_{xy_j}\left(t\right)} \right|^2}{\sum_{j=1}^{n} m_{x_j}\left(t\right)^2 m_{y_j}\left(t\right)^2}$$

where *n* is the number of trials. For the time point *t* in the trial *j*, $m_x\left(t\right)$ and $m_y\left(t\right)$ are the time-varying magnitude of the analytic signals from a MEG sensor (*x*) and the photodiode (*y*) respectively, $\phi_{xy}\left(t\right)$ is the phase difference as a function of time (for a detailed description, please see *Cohen, 2014*).

### Selection for the RIFT response sensors

MEG sensors that showed significantly stronger coherence at 60 Hz during the pre-target segments than the baseline segments were selected as the RIFT response sensors. We used a non-parametric Monte-Carlo method (*Maris et al., 2007*) to estimate the statistical significance. The pre-target segments were constructed by pooling the target contextual congruity conditions together. Several previous RIFT studies from our lab observed robust tagging responses from the visual cortex for flicker above 50 Hz (*Drijvers et al., 2021*; *Duecker et al., 2021*; *Zhigalov et al., 2019*; *Zhigalov and Jensen, 2020*). Thus, this sensor selection procedure was confined to the MEG sensors in the visual

cortex (52 planar sensors). Here, the pre-target segments and baseline segments were treated as two conditions. For each combination of the MEG sensor and photodiode channel, coherence at 60 Hz was estimated over trials for the pre-target and baseline conditions separately. Then, we calculated the z-statistic value for the coherence difference between pre-target and baseline using the following equation (for details please see *Maris et al., 2007*; *Equation 3*):

$$Z = \frac{(\tanh^{-1}(|\text{coh}_1|) - \text{bias}_1) - (\tanh^{-1}(|\text{coh}_2|) - \text{bias}_2)}{\sqrt{\text{bias}_1 + \text{bias}_2}}$$

$$\text{bias}_1 = \frac{1}{2n_1 - 2}, \text{bias}_2 = \frac{1}{2n_2 - 2}$$

where $\text{coh}_1$ and $\text{coh}_2$ denote the coherence value for the pre-target and baseline segments, $\text{bias}_1$, and $\text{bias}_2$ is the term used to correct for the bias from trial numbers of the pre-target ($n_1$) and baseline condition ($n_2$). All trials from the pre-target and baseline conditions were used.

After obtaining the z statistic value for the empirical coherence difference, we ran a permutation procedure to estimate the statistical significance of this comparison. We randomly shuffled the trial labels between pre-target and baseline conditions 5000 times. During each permutation, coherence was computed for both conditions (with shuffled labels), then entered *Equation 3* to obtain a z score for the coherence difference. After all randomizations were performed, the resulting z-values established the null distribution. Since a tagging response sensor was supposed to have stronger coherence during the pre-target segments compared with the baseline segments, the statistical test was right-sided. If the z-value of the empirical coherence difference was larger than 99% of z-values in the null distribution, this sensor was selected as the RIFT response sensor (right-sided, p=0.01). For each participant, the coherence values were averaged over all sensors with significant tagging response to obtain an averaged coherence for further analyses. Please note that the tagging response sensors may vary in number across participants (7.9±4.5 sensors per participant, M ± SD). Additionally, they may have a different but overlapping spatial layout, primarily over the visual cortex. For the topography of all tagging response sensors, please refer to *Figure 2A*.

## Coherence response curve

We first extracted MEG segments for the words N-4, N-3, N-2, N+1, N+2, and N+3 following the same procedure described in the pre-processing when extracted MEG segments for the pre-target (N-1) and target words (N). All segments were 1 s long, aligned with the first fixation onset to the word. Then, we calculated the coherence at 60 Hz during these segments for participants who have RIFT response sensors (n=29). Next for each participant, the 60 Hz coherence was first averaged over the RIFT response sensors, then averaged within a time window of [0 0.2] s (the averaged fixation duration for words). We got an averaged of 60 Hz coherence for the word at each position. We also got the 60 Hz coherence for the baseline interval averaged over [0 0.2] s, aligned with the cross-fixation onset. Then a pairwise *t*-test was performed between the baseline coherence and the coherence at each word position.

## Coherence comparison between conditions

The coherence comparison analyses were only conducted for the participants who had sensors with a reliable tagging response (n=29). To avoid any bias from trial numbers, an equal number of trials under the different contextual congruity conditions was entered into the coherence analysis per participant. We randomly discarded the redundant trials from the condition that had more trials for both the pre-target and target segments.

To compare the pre-target coherence amplitude between conditions, the coherence values at 60 Hz were averaged across the minimum fixation duration of all pre-target words. The time window for averaging was defined for each participant so that the coherence signal from the target fixation was not involved. Similarly, we averaged the 60 Hz coherence for the target segments over the minimum target fixation duration. Then, a two-sided pairwise Student's *t*-test was performed to estimate the statistical significance of the coherence difference as shown in *Figure 3B* and *Figure 4B*.

To assess the coherence onset latency difference between conditions, we used a leave-one-out Jackknife-based method (*Miller et al., 1998*). We extracted the 60 Hz coherence during the 1 s long

pre-target segments for each participant. Then, during each iteration of participants, we randomly chose and left out one participant. For the remaining participants, coherences were calculated for the incongruent and congruent target conditions. Then, the coherence was averaged over the remaining participants to estimate the onset latency for both conditions. Here, the onset latency was defined as the time point when the averaged coherence value reached its half-maximum $(\text{coh}_{\text{min}} + (\text{coh}_{\text{max}} - \text{coh}_{\text{min}})/2)$. We computed the onset latency difference by subtracting the onset latency for the incongruent target condition from the congruent condition. After all iterations, onset latency differences from all these subsamples were pooled together to estimate a standard error ($S_D$) using the following equation (***Equation 4***):

$$S_D = \sqrt{\frac{n-1}{n} \cdot \sum_{i=1}^{n} \left(D_{-i} - \bar{J}\right)^2}$$

where $\bar{J}$ is the average onset latency difference over all the subsamples, $D_{-i}$ is the coherence difference obtained from the subsample when participant $i$ was left out, $n$ is the number of participants. We also computed the onset latency difference from the overall sample set (without leaving any participant out) and divided it by the $S_D$ to obtain its $t$-value. A standard t table (pairwise, two-tailed) provided the statistical significance for the coherence onset latency difference between the incongruent and congruent target conditions. This procedure was conducted for both the pre-target and target segments as shown in ***Figure 3C*** and ***Figure 4C***.

## Source analysis for RIFT

We used a beamforming-based approach, DICS (***Gross et al., 2001***), to estimate the neural sources that generated the responses to RIFT. The DICS technique was applied to the pre-target segments (0–0.5 s aligned with fixation onset to the pre-target word) regardless of the target contextual congruity conditions, with a focus of 60 Hz in the frequency domain. In this source analysis, only participants with robust tagging responses were included (n=29).

First, we constructed a semi-realistic head model, where spherical harmonic functions were used to fit the brain surface (***Nolte, 2003***). We aligned the individual structural MRI image with the head shape that was digitised during the MEG session. This was done by spatially co-registering the three fiducial anatomical markers (nasion, left and right ear canal) and extra points that covered the whole scalp. For participants whose MRI image was unavailable, the MNI template brain was used instead. The aligned MRI image was segmented into a grid, which was used to prepare the single-shell head model.

Next, we constructed the individual source model by inverse-warping a 5 mm spaced regular grid in the MNI template space to each participant's segmented MRI image. We got the regular grid from the Fieldtrip template folder, which was constructed before doing the source analysis. In this way, the beamformer spatial filters were constructed on the regular grid that mapped to the MNI template space. Even though after this warping procedure grid points in the individual native space were no longer evenly spaced, the homologous grid points across participants were located at the same location in the normalised template space. Thus, the reconstructed sources can be directly averaged across participants on the group level.

Next, the Cross-Spectral Density (CSD) matrix was calculated at 60 Hz for both the pre-target and baseline segments. The CSD matrix was constructed for all possible combinations between the MEG sensors and the photodiode channel. No regularisation was performed to the CSD matrices (lambda = 0).

Finally, a common spatial filter was computed based on the individual single-shell head model, source model, and CSD matrices. This spatial filter was applied to both the pre-target and baseline CSD matrices for calculating the 60 Hz coherence. This was done by normalizing the magnitude of the summed CSD between the MEG sensor and the photodiode channel by their respective power. After the grand average over participants, the relative change for pre-target coherence was estimated as the following formula, $(\text{coh}_{\text{pretarget}} - \text{coh}_{\text{baseline}})/\text{coh}_{\text{baseline}}$.

## Acknowledgements

We thank Jonathan L Winter for providing help with the MEG recordings. The computations described in this paper were performed using the University of Birmingham's BlueBEAR HPC service, which provides a High Performance Computing service to the University's research community. See http://www.birmingham.ac.uk/bear for more details. This study was supported by a Leverhulme Early Career Fellowship awarded to YP (ECF-2023–626) and the following grants to OJ: the James S McDonnell Foundation Understanding Human Cognition Collaborative Award (grant number 220020448), Wellcome Trust Investigator Award in Science (grant number 207550), and the BBSRC grant (BB/R018723/1) as well as the Royal Society Wolfson Research Merit Award. The funders had no role in study design, data collection and analysis, decision to publish, or preparation of the manuscript.

## Additional information

### Funding

| Funder | Grant reference number | Author |
| --- | --- | --- |
| Leverhulme Trust | ECF-2023-626 | Yali Pan |
| Wellcome Trust | 10.35802/207550 | Ole Jensen |
| Biotechnology and Biological Sciences Research Council | BB/R018723/1 | Ole Jensen |
| Royal Society | Wolfson Research Merit Award | Ole Jensen |

The funders had no role in study design, data collection and interpretation, or the decision to submit the work for publication. For the purpose of Open Access, the authors have applied a CC BY public copyright license to any Author Accepted Manuscript version arising from this submission.

### Author contributions
Yali Pan, Conceptualization, Data curation, Formal analysis, Funding acquisition, Visualization, Methodology, Writing – original draft, Project administration, Writing – review and editing; Steven Frisson, Kara D Federmeier, Conceptualization, Writing – original draft, Writing – review and editing; Ole Jensen, Conceptualization, Supervision, Funding acquisition, Writing – original draft, Writing – review and editing

### Author ORCIDs
Yali Pan ⓘD http://orcid.org/0000-0003-0062-4326
Ole Jensen ⓘD http://orcid.org/0000-0001-8193-8348

### Ethics
Human subjects: The study was approved by the University of Birmingham Ethics Committee(under the approved Programme ERN_18-0226AP27). The informed consent form was signed by all participants after the nature and possible consequences of the studies were explained.

Reviewer #1 (Public Review): https://doi.org/10.7554/eLife.91327.4.sa1
Author response https://doi.org/10.7554/eLife.91327.4.sa2

## Additional files

### Supplementary files
• MDAR checklist

### Data availability
We have deposited the following data in the current study on figshare (https://figshare.com/projects/Semantic/149801): the epoch data after pre-processing, the raw EyeLink files, the Psychotoolbox

data, and the head models after the co-registration of T1 images with the MEG data. The experiment presentation scripts (Psychtoolbox), statistics scripts (R), scripts and data to generate all figures (Matlab) are available on GitHub (https://github.com/yalipan666/Semantic, copy archived at *Pan, 2022*).

The following datasets were generated:

| Author(s) | Year | Dataset title | Dataset URL | Database and Identifier |
|---|---|---|---|---|
| Pan Y, Frisson S, Federmeier KD, Jensen O | 2024 | Headmodels | https://doi.org/10.6084/m9.figshare.21618657.v2 | Figshare, 10.6084/m9.figshare.21618657.v2 |
| Pan Y, Frisson S, Federmeier KD, Jensen O | 2022 | epoch data after pre-processing | https://doi.org/10.6084/m9.figshare.21206990.v4 | Figshare, 10.6084/m9.figshare.21206990.v4 |
| Pan Y, Frisson S, Federmeier KD, Jensen O | 2022 | Experiment information | https://doi.org/10.6084/m9.figshare.21618708.v1 | Figshare, 10.6084/m9.figshare.21618708.v1 |
| Pan Y, Frisson S, Federmeier KD, Jensen O | 2022 | EyeLink files | https://doi.org/10.6084/m9.figshare.21618645.v1 | Figshare, 10.6084/m9.figshare.21618645.v1 |
| Pan Y, Frisson S, Federmeier KD, Jensen O | 2022 | Psychtoolbox data | https://doi.org/10.6084/m9.figshare.21618636.v1 | Figshare, 10.6084/m9.figshare.21618636.v1 |

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

# Appendix 1

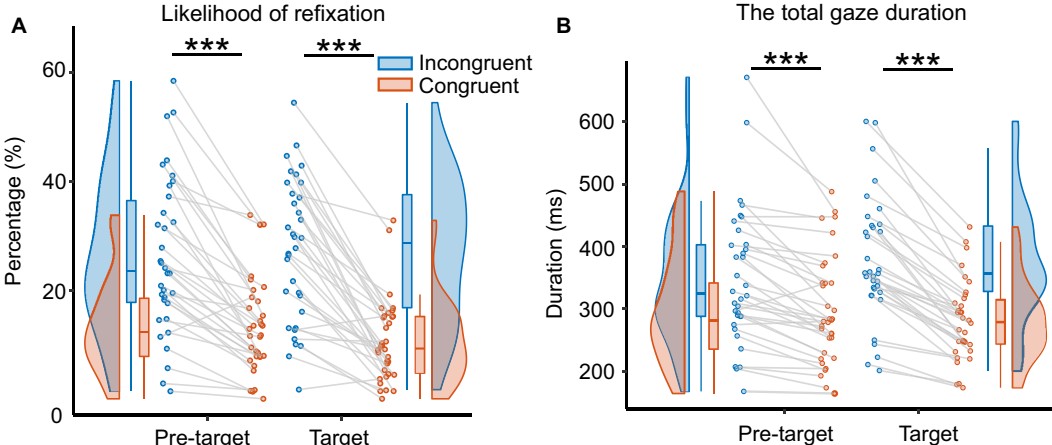

**Appendix 1—figure 1.** The likelihood of refixation and total gaze duration of eye movement data. (**A**) The likelihood of refixation into a word was defined as the proportion of trials that have at least one regression from a later part of the sentence back to that word. We found that when the target words were contextually incongruent with the sentence compared with congruent, there was a significantly higher probability of regression into pre-target words ($t_{(33)}$ = 7.83, p=5.04×10$^{-9}$, $d$=1.34, two-sided pairwise $t$-test) and target words ($t_{(33)}$ = 9.13, p=1.49×10$^{-10}$, $d$=1.57, two-sided pairwise $t$-test). Each dot indicates one participant. (**B**) The total gaze duration was defined as the sum of all fixations on a given word, including those fixations during re-reading. Significantly longer total gaze durations were found for pre-target words ($t_{(33)}$ = 5.78, p=1.86×10$^{-6}$, $d$=0.99, two-sided pairwise $t$-test) and target words ($t_{(33)}$ = 10.55, p=4.20×10$^{-12}$, $d$=1.81, two-sided pairwise $t$-test) when the target words were incongruent with the context compared with congruent. \*\*\*p<.001; n.s., not statistically significant.

## Experimental sentence set

### Here, we share all 160 sentences embedded with congruent target words

For sentence set version A, we swapped the target words within each pair for sentences 1–80 and made them incongruent, while sentences 81–160 were kept congruent. For sentence set version B, target words in sentences 1–80 were kept the same but target words in sentences 81–160 were swapped within each pair to make them incongruent. The sequence of the sentences was shuffled to make sure that no more than three sentences in a row were in the same condition. For illustration, the target words are shown in italic type here, but they were in normal type in the experiment. For the 117 filler sentences, please see the Appendix in *White, 2008*.

1. Last night, my lazy *brother/jacket* came to the party 1 min before it was over.
2. Lily says this blue *jacket/brother* will be a big fashion trend this fall.
3. This area has been populated by many *hikers/coins* over the last year.
4. Little Jimmy picked up a box and put some *coins/hikers* inside of it.
5. Joey became an avid *student/ring* during his adolescence.
6. He could only afford a cheap *ring/student* without a diamond for his fiancée.
7. This morning the noisy *kids/ideas* played happily in the backyard.
8. My parents had no firm *ideas/kids* about what I should become.
9. The unfortunate pupil lost his beloved *pony/crisis* just before his birthday.
10. Experts say that the severe *crisis/pony* will cause oil prices to triple.
11. The construction of this ancient *castle/worker* cost a lot of money.
12. After the meeting, the anxious *worker/castle* sighed in the hallway.
13. Peter's love for this sporting *match/collar* inspired all his friends.
14. We could see from her torn *collar/match* that she had been in a fight.
15. With the help of his clever *friend/burger* Jack, he made the first pot of gold.
16. I always like to order a filling *burger/friend* from the local pub.
17. She looked at the tired *fireman/scan* with a satisfied smile.

18. He felt relieved after completing the complex *scan/fireman* within an hour.
19. Scientists found a steep *boulder/cousin* sitting in the middle of the canyon.
20. Last week his friendly *cousin/boulder* passed out for no apparent reason.
21. They asked the selfish *maid/roof* where her huge sums of money came from.
22. It took Tom a month to mend the broken *roof/maid* all by himself.
23. Under stress, the crafty *boss/rifles* promised customers a full refund.
24. The cowboys hung the stolen *rifles/boss* high up on the wall.
25. She submitted the crucial *file/queen* that can prove her innocence.
26. According to history books, the proud *queen/file* never accepted any criticism.
27. Ana complained that the tall *herbs/sport* behind the house had dried up.
28. He failed in his chosen *sport/herbs* with hopes of success fading with each effort.
29. She said that the corrupt *company/bush* offered high salaries to young graduates.
30. In the last few years, the thick *bush/company* died back dramatically.
31. It turned out that the last- minute *trip/tree* lasted for 6 hr.
32. Linda found that the slender *tree/trip* dead from a pest infestation.
33. Suddenly, the warm *coffee/flower* stained his brand new shirt.
34. Plenty of rain will make the vivid *flower/coffee* blossom well.
35. Jack became a humble *chef/vehicle* specializing in French cuisine.
36. The young man's shiny *vehicle/chef* vanished slowly out of sight.
37. To the north, the steep *hills/colonel* stretched for many miles.
38. Before sleeping, the nervous *colonel/hills* smoked a cigarette.
39. Decades ago, that algae- covered *pond/player* was enough to irrigate the crops.
40. He saw the smart *player/pond* throw the ball, causing chaos among the opposition.
41. In recent days, the cruel *murder/cream* has scared citizens from going out.
42. Mary told me that the light *cream/murder* was low in fat but hard to whip.
43. News said that the painful *disease/ball* would continue to affect many children.
44. The boy found his lost *ball/disease* under the tree and stopped crying at once.
45. Politicians hated the brief *report/drone* criticizing the government's incompetence.
46. My favourite gift is the shiny *drone/report* from my dad last year.
47. Every year, the sandy *shore/officer* attracts thousands of tourists.
48. After taking a deep breath, the junior *officer/shore* entered the room.
49. Last week, the caring *family/plaza* rescued a stray dog and kept it as a pet.
50. During the air raid, the spacious public *plaza/family* happened to be ruined.
51. With his sharp criticism, the young *actor/storm* annoyed his agent as usual.
52. Laura was told that the sudden *storm/actor* delayed the bus for two days.
53. Lily said that the vacant *cottage/picture* belonged to her grandparents.
54. In the small house, a comic *picture/cottage* adorned the reception room.
55. Facing the lion, the brave *hunter/engine* showed no fear.
56. Out of repair, the rattling *engine/hunter* was about to be scrapped.
57. Jack had to admit that this planned *visit/aunt* turned out to be embarrassing.
58. Tom admired the way his devoted *aunt/visit* always volunteers on weekends.
59. Rob felt that the brief *letter/clerk* from his wife expressed a hint of sadness.
60. Alone at home, the tired *clerk/letter* cooked a beef patty.
61. After the surgery, Rob's poor *health/dusk* left him barely able to get out of bed.
62. Sam's train arrived before *dusk/health* and we were able to give him a ride home.
63. She gave the dog a quick *bath/joke* after they came back from the outside.
64. Michael made a mean *joke/bath* about Boris Johnson's hair.
65. They didn't realize the harsh *impact/crown* that their products could have.
66. In the museum, we saw the golden *crown/impact* that belonged to the first king.
67. Bill is a superb *partner/night* because he is easy to get along with.
68. The explorer made his way through the gloomy *night/partner* with a small torch.
69. The TV show was an obvious *flop/lady* after the actress joined the cast.
70. On rainy days, the careful *lady/flop* reminded herself to go slowly.
71. Jane complained that her white *kitten/problem* hadn't come home for two days.
72. I guess no one can solve the hard *problem/kitten* without outside help.

73. The new event was such a huge *failure/suspect* that people kept talking about it.
74. Before committing the crime, the anxious *suspect/failure* drank a lot of alcohol.
75. Toby kept his money in a small *shed/deer* because he lived on a farm.
76. They noticed the young *deer/shed* eating acorns in the forest.
77. Amy wanted some more of the sliced *pear/canal* for afternoon snack.
78. Laura went down to the narrow *canal/pear* to watch the boats.
79. I wondered if the noisy *club/pain* would be a good place for the bachelorette party.
80. Tara always has an acute *pain/club* in her tooth after eating ice cream.
81. Before the war, the brave *general/potato* assembled an army.
82. She began to slice up a large *potato/general* for the dinner.
83. The child had a large *face/mist* with big, expressive eyes.
84. Last night, there was a dense *mist/face* when they left the cinema.
85. Marla enjoyed seeing the chubby *cats/court* playing with each other.
86. Jim entered the giant *court/cats* to try out for the basketball team.
87. They visited the antique *chapel/fans* before booking their wedding.
88. After the defeat, the crazy *fans/chapel* kept cursing and crying.
89. She approached the rusty *gate/toast* before realizing it was locked.
90. Many people like to eat crispy *toast/gate* with their morning coffee at breakfast.
91. They stepped into the messy *garage/hawk* that had high wooden shelves.
92. We watched the large hungry *hawk/garage* swoop down to get the poor chicken.
93. Alexandra used a short *hammer/museum* when she created the stone statue.
94. Ruth visited the public *museum/hammer* that she had read about all these years.
95. We'd better buy some tasty *chips/speech* before we watch the big game.
96. Historians believe the rousing *speech/speech* heralded the start of the revolution.
97. Under the tree, there is a little *hare/moon* running happily.
98. In the darkness, only the misty *moon/hare* lit up the street.
99. The prince inherited the supreme *power/sheep* from the late king.
100. Look over there, a fluffy *sheep/power* seems to be lost.
101. The man was a young *teacher/opinion* who always worked late into the night.
102. As for this scandal, Jo has a clear *opinion/teacher* but she won't say it.
103. They had no idea that the blue *liquid/justice* shrinks all woollen clothes.
104. The report was sent to the honest *justice/liquid* three days before the trial.
105. Every night, this tired *captain/bottle* drank wine before going to sleep.
106. The shopkeeper said the metal *bottle/captain* would sell well this year.
107. For the locals, the salt *lake/animal* triggered a political issue.
108. Near the small brook, a hungry *animal/lake* hunts quietly for hours.
109. Every night, this deep *secret/surgeon* makes the pianist toss and turn.
110. In the lab, a young *surgeon/secret* examined the victim's body.
111. Sadly, the lonely *poet/flour* died before he could finish his last poem.
112. Due to the moist weather, the wheat *flour/poet* became mouldy quickly.
113. Sue's colleagues say that her warm *heart/screen* makes everyone like her.
114. On the wall, the small green *screen/heart* shows the room temperature precisely.
115. To his surprise, the yummy *dish/nanny* was not expensive.
116. Eventually, the greedy *nanny/dish* disclosed all the details about this affair.
117. The sight of the cotton *factory/patient* was something to behold.
118. It was obvious that the weak *patient/factory* was getting weaker day by day.
119. In the past month alone, the gentle *scholar/pots* published five papers.
120. The filthy and rusty *pots/scholar* made the food taste terrible.
121. Just after dawn, an armed *ship/shirt* approached the pretty lagoon slowly.
122. At last, she found the wool *skirt/ship* hanging in the wardrobe.
123. Villagers said that the newly built *school/crowd* was well equipped.
124. In the downtown market, the agitated *crowd/school* began the parade.
125. Suzy really likes eating *sugar/music* because she wasn't allowed to eat it as a kid.
126. Ali said he really enjoyed modern *music/sugar* when he was at college.
127. Tina wants a spacious *yard/chief* because she likes to lie on the grass and read.

128. In an open field, the violent *chief/yard* executed prisoners with a gun.
129. After working overtime for a month, the wronged *manager/grass* wanted to jump ship.
130. In the Stone Age, the spiny *grass/manager* prevailed over the land.
131. The sparrow was being chased by some fluffy *hens/cups* under the hot sun.
132. When the ball was scored, they tapped their *cups/hens* to show their joy.
133. They recorded the details of the stolen *cars/legs* carefully on a spreadsheet.
134. The poor boy stood in the snow with bruised *legs/cars* and cried sadly.
135. Little Roy likes to play with the plastic *bricks/lawyer* at the Lego store.
136. It was said that the honest *lawyer/bricks* convened the committee meeting.
137. I learned about the muddy *trail/jury* through a friend on the last hike.
138. Just now, the calm *jury/trail* delivered a guilty verdict in this notorious case.
139. The holiday was neglected by this busy *parent/story* but her son was used to it.
140. This widely spread *story/parent* reflected the distortion of human nature.
141. We are meeting at the newly built *airport/editor* tonight for our trip to Europe.
142. Every day before leaving work, the tall *editor/airport* cleans her desk.
143. Nobody knew when the excited *puppy/area* urinated on the floor.
144. Everyone knows that entire *area/puppy* has restricted access.
145. The man's cunning *excuse/truck* relieved him of the fine.
146. Roy repaired the broken *truck/excuse* over the weekend.
147. Ana was glad that the gentle *nurse/meeting* said her little boy was out of danger.
148. In the company, the annual *meeting/nurse* marks the end of a year's hard work.
149. He carefully placed the sharp *sword/desk* down after the fight.
150. She found an empty *desk/sword* where she could put her computer.
151. They danced a slow *tango/note* together after dinner.
152. Steph noticed a torn *note/tango* and looked for the other half.
153. Mindy's dog has a strange *smell/tape* and likes to bark a lot.
154. Patty likes to cut some pink *tape/smell* to decorate her notebooks.
155. We could hear the angry *priest/card* shouting at the little girl.
156. David was happy to receive a nice *card/priest* from his daughter at Christmas.
157. She always meets the same happy *couple/hole* when she walks in the park.
158. The stray dog lives in a hidden *hole/couple* that protects it from the cold weather.
159. Jack failed to submit his concise *paper/baker* before the deadline.
160. I have heard that the young *baker/paper* makes the best baguettes in town.

