## [Editor Report · eLife assessment]

This **important** study contributes to the understanding of how parafoveal words are neurally processed during naturalistic sentence reading. **Convincing** evidence is provided that the MEG response to a word can be modulated by the semantic congruency of a parafoveal target word. The study addresses a classic question in reading using a new Rapid Invisible Frequency Tagging (RIFT) technique, which can separately monitor the neural processing of multiple words during sentence reading.

---

## [Referee Report · Reviewer #1 (Public Review)]

Summary:

The study investigates parafoveal processing during natural reading, combining eye-tracking and MEG techniques, building upon the RIFT paradigm previously introduced by Pan et al. (2021).

The manuscript is well-written with a clear structure, and the data analysis and experimental results are presented in a lucid manner.

Comments on revised version:

I am satisfied with the revisions made by the authors. I believe the study introduces a new research paradigm to the field.

---

## [Author Response]

The following is the authors’ response to the previous reviews.

**Public Reviews:**

**Reviewer #1 (Public Review):**
The study investigates parafoveal processing during natural reading, combining eye-tracking and MEG techniques, building upon the RIFT paradigm previously introduced by Pan et al. (2021). Overall, the manuscript is well-written with a clear structure, and the data analysis and experimental results are presented in a lucid manner.The authors have addressed the issues I raised in the previous round of review to my satisfaction. However, I still have two concerns that require the authors' consideration.Firstly, the similarity between the RIFT analysis process in this study and traditional ERP analysis could lead readers to equate RIFT with components like N400, potentially influencing their interpretation of the results. Although the author's response has somewhat clarified my queries, I seek confirmation: does RIFT itself signify "visual attention" or the "allocation of attentional resources to the flickering target words" (line 208) in this study? While this may not be pivotal, as it primarily serves as an indicator to evaluate whether contextual congruity can indeed modulate the RIFT response rather than indicating early parafoveal semantic integration, I recommend that the authors explicitly address this point in the manuscript, maybe in the discussion section, to enhance reader comprehension of the article's rationale.Secondly, regarding the study's conclusions, there appears to be an overemphasis in stating that "semantic information ... can also be integrated with the sentence context ..." (line 21-22). As raised by Reviewer 2 (Major Point 1) and acknowledged by the authors in the limitations of the revised manuscript (lines 403-412), the RIFT effect observed likely stems from local congruency. Therefore, adjusting the conclusion to "integrated with previous context" may offer a more precise reflection of the findings.

We appreciate the positive comments from the Reviewer.

In response to the first concern, we have rephrased the sentence (Line 207-209 in the revised manuscript) to clarify that RIFT measure visual attention : “Moreover, as RIFT directly measures visual attention, the left-skewed RIFT response curve suggests that more visual attention is allocated towards the flickering target words before fixating on them, aligning with the left-to-right order of reading English.”

Regarding the second concern, we have addressed the issue by modifying “sentence context” to “previous context” in both the Abstract (Line 18 and Line 22) and the Discussion section (Line 314 and Line 361) of the revised manuscript.